# The Latest Evidence of Erythropoietin in the Treatment of Glaucoma

**DOI:** 10.3390/ijms232416038

**Published:** 2022-12-16

**Authors:** Ting-Yi Lin, Yi-Fen Lai, Yi-Hao Chen, Da-Wen Lu

**Affiliations:** Department of Ophthalmology, Tri-Service General Hospital, National Defense Medical Center, Taipei 11490, Taiwan

**Keywords:** erythropoietin, glaucoma, neuroprotection, optic neuropathy, retinal ganglion cell

## Abstract

Erythropoietin (EPO) is a circulating hormone conventionally considered to be responsible for erythropoiesis. In addition to facilitating red blood cell production, EPO has pluripotent potential, such as for cognition improvement, neurogenesis, and anti-fibrotic, anti-apoptotic, anti-oxidative, and anti-inflammatory effects. In human retinal tissues, EPO receptors (EPORs) are expressed in the photoreceptor cells, retinal pigment epithelium, and retinal ganglion cell layer. Studies have suggested its potential therapeutic effects in many neurodegenerative diseases, including glaucoma. In this review, we discuss the correlation between glaucoma and EPO, physiology and potential neuroprotective function of the EPO/EPOR system, and latest evidence for the treatment of glaucoma with EPO.

## 1. Introduction and the Relation between Glaucoma and EPO

Glaucoma, the second leading cause of blindness worldwide, is a progressive and multifactorial optic neuropathy characterized by the loss of retinal ganglion cells (RGCs). The number of patients with glaucoma worldwide is estimated to reach 111.8 million by 2040, with an especially higher impact in Asia and Africa [1]. Recently, glaucoma is regarded as a neurodegenerative disorder involving the eye and brain with some common traits with Alzheimer’s disease (AD) and other tauopathies [2]. According to population-based studies, the most common risk factors of glaucoma are elevated intraocular pressure (IOP), old age, reduced ocular perfusion pressure, high myopia, a family history of glaucoma, and ethnic and genetic susceptibilities [3,4]. Elevated IOP is the major risk factor and therapeutic target for glaucoma. However, the IOP of patients with glaucomatous optic neuropathy and associated visual field loss may be within the statistically normal range, suggesting the existence of different underlying pathogenic mechanisms. As a result, some glaucoma researchers endeavored to discover other pathogeneses related to RGC death in glaucoma and alternative therapeutic modalities other than IOP-lowering agents.

Erythropoietin (EPO), a hematopoietic cytokine, has been studied for its potential therapeutic effects in many neurodegenerative diseases, including Parkinson’s disease, AD, amyotrophic lateral sclerosis, spinal cord injury, brain ischemia, hypoxia, and hyperoxia [5]. In ophthalmology, the EPO receptor (EPOR) is expressed in human retinal tissues, such as the photoreceptor cells, retinal pigment epithelium, and RGC layer [6,7]. Emerging evidence has suggested a potent neuroprotective effect of EPO in the brain and retina [5,8,9]. In cultured retinal neurocytes, it promotes neurite outgrowth in a dose-dependent manner. Additionally, it increases the survival and decreases the apoptosis of retinal neurocytes from glutamate-induced cytotoxicity [10]. Tezel et al. showed a two-fold upregulation of the hemoglobin (Hb) expression in ocular hypertensive rat eyes and glaucomatous human donor eyes. The hypoxia-inducible EPO signaling was found to regulate the expression of Hb in the macroglia and RGCs and was suggested to be an intrinsic protective mechanism of neuronal cells [11]. Previous studies have also found a significantly higher aqueous levels of EPO in primary open-angle glaucoma, acute and chronic primary angle-closure glaucoma, neovascular glaucoma, and pseudoexfoliative glaucoma [12,13,14,15]. Tissue hypoxia, the major stimulating factor of EPO production, was also found in the retina and optic nerve head of glaucomatous eyes, and hypoxic signaling transduction is considered to be a likely component of the pathogenic mechanisms of glaucomatous neurodegeneration [16]. Consequently, there is a growing interest in the study of the potential therapeutic effects of EPO in glaucoma patients. In this review, we introduce and discuss the neuroprotective function of the EPO/EPOR system and the latest evidence of the treatment of glaucoma with EPO.

## 2. EPO/EPOR System and Its Potential Neuroprotective Function

Human EPO comprises 165 amino acids, mainly produced by the liver of the fetus and newborn, and by the kidney soon after birth [17]. EPO is a circulating hormone that stimulates erythropoiesis by promoting the maturation and survival of erythroid progenitors [18]. In adults, renal interstitial fibroblasts account for approximately 90% of the overall EPO production [19]. The gene expression of EPO is induced by anemia or hypoxia and regulated by oxygen tension through a transcriptional feedback loop [18,20,21].

The hypoxia-inducible factor (HIF) plays a key role in the hypoxic response by modulating downstream genes that regulate vital biological processes, such as angiogenesis, cell survival and proliferation, metabolism, erythropoiesis, and cell cycle progression [22]. Functional HIF transcription factors are heterodimers containing either types of the oxygen-labile α-subunit, HIF1-α or HIF2-α, and a stable β-subunit [23]. The stability of HIF-α is regulated by 2-oxoglutarate-dependent oxygenases, prolyl hydroxylase domain (PHD) proteins, which catalyze the hydroxylation of specific proline residues in the HIF-α subunit. The hydroxylated HIF-α is subsequently ubiquitylated by von Hippel–Lindau tumor suppressor-E3 ubiquitin ligase complex and rapidly degraded by the proteasome [21,24,25,26]. Under low oxygen level conditions, the function of PHD proteins is inactive, and functional heterodimers of HIF are formed. Furthermore, the protein factor inhibiting HIF-1 (FIH) also participates in regulating the transcriptional activity of HIF. Under normoxic conditions, FIH suppresses the activity of HIF-α C-terminal transactivation domain (CAD) via hydroxylation and directly competes with the p300 transcription coactivator for CAD binding [27] (Figure 1).

The EPOR is a classical type-I cytokine receptor composed of an extracellular domain of 225 amino acids, a transmembrane region of 23 amino acids, and a cytoplasmic domain of 235 amino acids [28,29]. The binding of EPO to the EPOR induces a dimerization and/or reorientation of EPOR monomers within a dimeric receptor structure, which then activate multiple signaling cascades including Janus tyrosine kinase 2 (JAK 2), signal transducer and activator of transcription 5 (STAT 5), mitogen-activated protein kinase (MAPK), extracellular signal-regulated kinases (ERK), phosphoinositide-3-kinase (PI-3-K)/Akt, and nuclear factor-kappa B (NF-kB) [30,31]. After binding with EPOR, EPO promotes erythropoiesis by maintaining the survival and facilitating the maturation of the colony-forming unit-erythroid progenitors and proerythroblasts [32,33]. 

Previous studies have demonstrated EPOR expression in multiple extra-hematopoietic tissues, including nervous system, skeletal muscle myoblasts, cardiomyocytes, endothelial cells, solid tumors, the liver, the uterus, and the retina [34,35,36,37]. In addition to facilitating red blood cell production, EPO possesses pluripotent potential, such as for cognition improvement, neurogenesis, and anti-fibrotic, anti-apoptotic, anti-oxidative, and anti-inflammatory effects [38,39,40,41]. The diverse nonhematopoietic activity of EPO in different tissues could be partly explained by the different isoforms of EPOR. Several isoforms of EPOR have been identified in previous studies, such as canonical homodimer isoform (EPOR_2_), EPOR/beta common receptor isoform (EPOR/βcR), substantia nigra dopaminergic isoform, soluble isoform (sEPOR), and EPO-producing hepatocellular receptor B4 [5,42,43]. Unlike the homodimer EPOR_2_, which is mainly involved in erythropoiesis, the EPOR/βcR heterodimer is suggested to be responsible for tissue protective effects [44,45]. An experiment involving cardiomyocytes and spinal cord injury models in βcR-knockout mice further supported the concept that EPOR/βcR comprises a tissue-protective heteroreceptor [46]. In retinal tissues, the EPOR/βcR heterodimer complex were found to be expressed in the RGC’s inner nuclear layer, and photoreceptor cells, through an in situ hybridization analysis of retinal sections [47]. sEPOR, on the contrary, was thought to be an endogenous antagonist of EPO, which competes with other EPOR isoforms to bind EPO and thus blocks its neuroprotective effects [48]. 

## 3. Therapeutic Effect of EPO in Glaucoma/RGC Survival Studies

Glaucoma is characterized by progressive death of RGCs and subsequent irreversible visual field loss. Owing to its neuroprotective potential, researchers have explored the therapeutic effects of EPO in optic neuropathies, particularly glaucoma. It has been proven that systemically applied EPO is able to cross the blood–brain barrier and blood–retinal barrier to exert its neuroprotective action on the central nervous system and eye [49,50]. Therefore, research focusing on the therapeutic effect of EPO in glaucoma has been studied both locally and systemically. Here, we review the latest evidence in the literature of the therapeutic effect of EPO in glaucomatous optic neuropathy and RGC death induced by different injury models (Table 1).

### 3.1. Mechanical Optic Nerve Injury

Experimental models of mechanical trauma of the optic nerve were used in many studies. Weishaupt et al. were the first to investigate the anti-apoptotic effects of recombinant human EPO (rhEPO) in rats with an axotomized optic nerve and in cultured immunopurified RGCs deprived of neurotrophin. The neuroprotective function of rhEPO was found to follow a bell-shaped dose–response curve both in vitro and in vivo without toxicity even at the highest concentration tested (8 U/eye), which suggests that EPO is a promising therapeutic molecule against neuronal apoptosis in glaucoma and neurodegenerative diseases. The investigation of possible intracellular signaling pathways mediating EPO neuroprotection revealed that the PI3K/Akt pathway was activated and subsequently inhibited the function of pro-apoptotic caspase-3 [51]. Another study using the transgenic mouse line tg21, which constitutively expresses human EPO in neuronal cells, showed that RGCs were protected against degeneration upon axotomy compared to wild-type control mice. An opposite result, however, was concluded by the authors that the neuroprotective effects of EPO critically depend on the activation of the ERK-1/-2 pathway but not on the Akt pathway [52]. This disparity among studies may be explained by differences in the species of animal models. 

King et al. found that a single intravitreal injection (IVI) of EPO not only increased the RGC somata and axon survival but also doubled the number of RGC axons regenerating along the peripheral nerve grafted onto the retrobulbar optic nerve 4 weeks after optic nerve transection. The JAK2/STAT3 pathway was delineated to be involved in the regenerative response by EPO [53]. By upregulating the expression of growth associated protein-43, EPO administered intravitreally also protected injured RGCs and promoted axonal regeneration after optic nerve crush injury [55]. An experiment involving an intraperitoneal injection of sustained release EPO-loaded poly lactic-co-glycolic acid/poly lactic acid (PLGA/PLA) microspheres in optic nerve crush rats was carried out by Rong et al. A single injection of EPO-loaded microspheres significantly prevented RGC death from neurodegenerative central nervous system diseases, such as glaucoma, with the same effectiveness as multiple injections of EPO [54]. 

### 3.2. Episcleral Vessel Cautery-Induced Glaucoma Model

A rat model of glaucoma induced using episcleral vessel cautery (EVC) was used to evaluate the potential neuroprotective effect of an IVI of recombinant rat EPO by Tsai et al., who found that the RGC counts in retinal specimens were significantly decreased in the EVC with and without intravitreal normal saline injection groups compared to unoperated control rats. However, the RGC counts did not differ significantly in the EVC with intravitreal EPO treatment group with the controls. Furthermore, the rhodamine-concanavalin A staining showed no evident adverse effects on the retinal vasculature at 21 days in the EPO-treated eyes. As a result, they concluded that a single IVI of 200 ng EPO exerted protective effects on RGC viability in an in vivo rat model of glaucoma [56]. 

Fu et al. studied the effect of exogenous rhEPO (Epoetin alpha) on the survival of RGC administered both intravitreally and systemically in a chronic ocular hypertension rat model induced by Argon laser photocoagulation of the episcleral and limbal veins. Two weeks after ocular hypertension, the levels of EPO protein and EPOR significantly increased by 1.5 and 3 times compared to normal retina, respectively. Additionally, neutralization of endogenous EPO with sEPOR exacerbated ocular hypertension-related RGC loss. RGC survival was significantly enhanced in both rhEPO treatment groups (IVI and intraperitoneal injection) compared with control eyes at 2 weeks. The study indicated that exogenous rhEPO could rescue RGCs and that an intrinsic recovery mechanism involving the endogenous EPO/EPOR system existed in chronic ocular hypertension [57].

Using a glaucomatous rat model induced by coagulation of the episcleral veins, Resende et al. confirmed that a subconjunctival administration of 1000 IU rhEPO could reach the ganglion cell layer (GCL) and last for at least 14 days after the administration [68]. A further study with a similar design showed that subconjunctival EPO administration exerted beneficial effects on retina both structurally and functionally in induced glaucoma in Wistar albino rats [58].

### 3.3. Model of Anterior Chamber Cannulation and IOP Elevation

Junk et al. cannulated the anterior chamber in rats and elevated IOP to 120 mmHg to evaluate the neuroprotective effects of EPO in a transient global retinal ischemia model. The expression of EPOR had increased significantly at 72 h. In addition, IVI of sEPOR had reduced the electroretinography (ERG) b-wave in a dose-dependent manner after 7 days of reperfusion. These findings supported that the endogenous EPO/EPOR system might participate in intrinsic protective mechanisms after acute ocular hypertension with retinal ischemia and that neurons might be rescued by exogenous application of EPO. Terminal deoxynucleotidyl transferase-mediated dUTP nick-end labeling (TUNEL) was used in the study as an indicator for neuronal cell apoptosis. Compared with the vehicle-treated group, the rhEPO-treated group resulted in significantly fewer TUNEL-positive cells in the GCL and inner nuclear layer, suggesting that rhEPO affords protection by inhibiting apoptosis. Results on histopathology, ERG, and TUNEL further implied that the systemic administration of rhEPO before or immediately after retinal ischemia had marked neuronal preservation in acute ischemic injury via an antiapoptotic mechanism [9]. 

A study using similar settings by Zhong et al. investigated the retrobulbar administration of rhEPO in acute elevated IOP rats. They cannulated the anterior chamber and raised the IOP to 70 mmHg for a duration of up to 60 min. Immediately afterwards, either 1000 units of rhEPO or a vehicle solution was administered via a retrobulbar injection. The number of surviving RGCs was significantly higher in the eyes with acute elevated IOP with rhEPO retrobulbar injection than in the eyes with acute elevated IOP with or without a vehicle solution retrobulbar injection. It was stated that EPO with a retrobulbar administration could protect RGCs from acute elevated IOP [59]. Jehle et al. assessed the function of EPO in the survival of RGCs in rat models of elevated IOP-induced ischemia and optic nerve compression. The IVI of 20 U EPO significantly increased the survival of RGCs in both models. EPO treatment significantly improved postischemic amplitudes of the a- and b-waves on ERG. The toxicological safety of EPO was confirmed by the result that neither the parameters of ERG and visual evoked potential (VEP), nor the number of RGCs differed significantly after IVI of EPO than in normal eyes [60].

### 3.4. Oligemia by Bilateral Common Carotid Artery Occlusion (BCCAO)

In a study involving a global oligemia rat model with BCCAO, the RGCs in rats receiving IVI of EPO were partially preserved. The neuroprotective effect of EPO on oligemic/ischemic retinas was suggested to related to the down-modulation of glial reactivity usually observed in hypoxic conditions, such as normal-tension glaucoma and retinal ischemia [61]. Another study using BCCAO-induced chronic cerebral ischemia rat model was conducted by Zhou et al. Three days following the procedure, the Sprague–Dawley rats were treated with rhEPO (50 U/100 g/week) or saline via intranasal administration for 8 weeks. On flash VEP results, the rhEPO treatment group significantly reduced the latency and elevated the amplitude of the P1 wave compared with the saline treatment group and showed no difference compared with the sham-operated rats. Similarly, the rhEPO-treated rats showed significant improvement in the thickness of the cerebral cortex and retina, number of neurons in the hippocampus CA1 region, and number of RGCs, implicating that rhEPO had strong neuroprotective effects over ischemic injury [62].

### 3.5. Mouse Strain Prone to Develop Glaucoma

Zhong et al., in their study using the DBA/2J mouse model of glaucoma, found no RGC loss up to 12 months of age of the mice treated with an intraperitoneal injection of various doses of EPO (3000, 6000, and 12,000 U/kg/week) compared to a 70% of total RGC loss in untreated control mice. Without affecting the IOP level, EPO promoted RGC survival in DBA/2J glaucomatous mice, indicating EPO as a potential therapeutic neuroprotectant in glaucoma [63].

### 3.6. Molecule-Induced Retinal Toxicity

Yamasaki et al. found that EPO protects RGCs from glutamate and nitric oxide-induced cytotoxicity and reverses the Bcl-2 expression in a dose-dependent manner in primary cultured RGCs from Wistar rats. The study suggested that EPO may be a potent neuroprotective agent for treating ocular diseases characterized by RGC death [64]. In our previous study investigating neuroprotective effect of EPO in the presence of N-methyl-d-aspartate (NMDA)-, trophic factor withdrawal-, and tumor necrosis factor-alpha-induced toxicity on cultured adult rat RGCs, EPO provided different degrees of neuroprotective effect depending on the type of neurotoxicity, RGC cell size, and the timing of treatment. Moreover, the signaling pathways of STAT-5, MAPK/ERK, and PI3K/Akt were found to play a part in the neuroprotective effect of EPO [65]. Recently, we conducted another study investigating the neuroprotective effects of EPO on NMDA-mediated RGC death in Wistar rats. An IVI of 50 ng EPO showed a significantly higher survival rate of the total, medium, and small RGCs after NMDA-mediated neurotoxicity. Additionally, EPO co-administered with NMDA downregulated the generation of pro-apoptotic proteins, such as μ-calpain, caspase 9, and Bax in GCL. Apart from its protective effects on RGCs, EPO reversed the damage to bipolar cell axon terminals in the inner plexiform layer induced by NMDA [66].

### 3.7. Microbeads-Induced Glaucoma Model

EPO-R76E, a mutant form of EPO to attenuated erythropoietic side effects, was used to load into either PLGA particles or poly(propylene sulfide) (PPS) microspheres in a recent animal study. Rex et al. injected either one of the two treatment materials intravitreally into a mouse model of microbeads-induced glaucoma. Both PLGA-EPO-R76E and PPS-EPO-R76E particles exerted neuroprotective effects through activation of the nuclear factor erythroid 2-related factor 2/antioxidant response element (NRF2/ARE) pathway [67].

## 4. Discussion

Various in vitro and in vivo animal models have been used to explore the potential therapeutic effects of EPO in glaucoma or RGC survival. However, there is still no well-established animal model that can perfectly mimic the human disease course. Owing to the diversity of the experiment design, animal species used, injury models, timing and route of drug administration, and type of EPO used in these studies, there is still no consensus on the treatment of EPO in glaucomatous optic neuropathy. As aforementioned, the exogenous EPO treatment was proven to promote RGC survival and preserve visual functions in different animal injury models and various modalities of EPO administration. The upregulated expressions of both EPO and EPOR after injury suggested the activation of an endogenous and compensatory response to promote neuronal survival [59,69]. In addition, results of sEPOR further confirmed the potential protective mechanisms of both the endogenous EPO/EPOR system and exogenous EPO treatment [57,70]. Although the potential benefits of exogenous EPO in glaucoma have been verified in many animal studies, there are no randomized controlled trials in the literature investigating the possible therapeutic effect of EPO in humans with glaucoma. In a study involving patients with chronic renal failure (CRF) undergoing peritoneal dialysis, the retinal nerve fiber layer (RNFL) thickness was significantly lower compared to normal controls. Compared to patients with CRF receiving systemic EPO treatment, those without treatment had a statistically lower RNFL thickness in the temporal quadrant [71].

Notably, treatment with EPO is associated with multiple adverse events, such as hypertension, nausea, headache, thromboembolic events, polycythemia, intracerebral hemorrhage, brain edema, and death, particularly when administered systemically and chronically or at a higher dose [72,73]. Therefore, researchers have endeavored to develop non-erythropoietic variants of EPO to dissociate the erythropoietic effect from the tissue protective effect through modification of EPO structures [74]. In our review of animal studies concerning the local or systemic administration of EPO in glaucoma, no significant adverse events were reported [51,53,56,57,60].

When applying medications for ocular diseases, it could be delivered via the local or systemic route of administration. Local administration is more favorable for ophthalmologists because of its lesser invasiveness, higher convenience, and minimized possible systemic adverse effects of the therapeutic agents. There are multiple applicable methods of local drug delivery in the eye including eye drops, and subconjunctival, subtenon, peribulbar, retrobulbar, intrastromal, intracameral, intravitreal, and subretinal injections. Local administration of EPO in the eyes theoretically decreases the possibility of systemic side effects; however, this requires further long-term studies for verification. 

Clinically applying EPO treatment in patients with glaucoma is hindered by safety concerns of its pro-angiogenic activity, as EPO was found to participate in some pathological conditions. EPO promotes angiogenesis by directly interacting with endothelial cells to stimulate their proliferation and migration and indirectly modulating the release of other angiogenic factors [75,76]. In proliferative diabetic retinopathy, locally increased EPO in the vitreous fluid acts independently of the vascular endothelial growth factor (VEGF) to induce retinal angiogenesis [77]. Concerning its role in tumor angiogenesis, EPO facilitates tumor cell survival, tumor growth, angiogenesis, and lymphangiogenesis through EPOR-dependent and EPOR-independent mechanisms [76]. In an experiment using a mouse model of oxygen-induced retinopathy (OIR), exogenous rhEPO promoted retinal neovascularization in a dose-dependent manner accompanied by elevated VEGF, endothelial nitric oxide synthase (NOS), and neuronal NOS expressions [78]. The hypothesis that EPOR signaling plays a role in the regrowth of vascularization following oxygen-induced capillary dropout and intravitreal angiogenesis was also suggested [37]. Evidence has showed an interaction between EPOR and VEGF receptor-2 to exacerbate STAT3 activation and lead to pathological angiogenesis in the setting of increased VEGFA [79]. Meanwhile, the timing of intervention is another significant parameter in EPO treatment. Early in the disease course of OIR, treatment with exogenous EPO prevented vessel dropout and subsequent hypoxia-induced neovascularization and protected against hypoxia-induced retinal neuron apoptosis. In contrast, exogenous EPO treatment in the late neovascular phase would rather enhance the severity of pathological neovascularization [80].

Despite not being fully understood yet, several mechanisms have been proposed to explain the neuroprotective function of EPO. EPO may inhibit apoptosis of neuronal cells by activation of the STAT5, MAPK, PI3K/Akt, and NF-κB signaling pathways, upregulation of anti-apoptotic factors such as Bcl-xL, and Bcl-2, and inhibition of the μ-calpain, Bax, caspase 8, and caspase 9 activities [64,66,70,81,82,83]. In chronic ocular hypertension, EPO was also found to inhibit the activation of HIF-1α to avoid neurotoxicity caused by oversynthesis of inducible NOS [84]. As such, EPO may protect against cell apoptosis through prevention of membrane phosphatidylserine exposure and genomic DNA destruction, inhibition of caspase 1, 3, 8, and 9 activities, blocking activation and proliferation of microglial cells, maintaining mitochondrial membrane potential, and preventing the release of cytochrome c through the activation of Akt pathway [85,86]. In addition to directly interacting with EPOR on RGCs, EPO may indirectly promote optic nerve survival by affecting the surrounding retinal cells, such as astrocytes, and Müller cells [87]. Furthermore, EPO may exert neuroprotective effects by indirectly affecting blood flow and cerebrospinal fluid pressure, and subsequently influence the translaminar pressure difference across the optic nerve head [88]. Of note, studies have revealed that EPO exerts its neuroprotective effects through mechanisms other than affecting IOP or inducing structural changes in aqueous outflow pathways [57,63]. 

New strategies of EPO therapy, such as injection of EPO gene encoding recombinant adeno-associated viral vector, and EPO-expressing mesenchymal stem cells transplantation, have been studied in ocular diseases [89,90]. Promising results on the neuroprotection and visual function preservation in glaucomatous animal models were found [88,91,92]. Nevertheless, significant technical hurdles and long-term safety of these novel therapies require further investigations. In the future, a safe and well-regulated treatment system will potentially carry forward the stem cell and gene therapies for glaucoma patients. Technological advancements, such as next-generation sequencing, would expectedly broaden our understanding of the underlying mechanisms of both the disease course of glaucoma and EPO signaling pathways involved in its therapeutic effects.

## 5. Conclusions

In conclusion, EPO and its derivatives continue to be a promising and novel treatment modality for glaucoma other than IOP-lowering agents. However, the underlying mechanisms of EPO-related neuroprotection still remain not fully understood. Most of the animal studies in this review utilized injury models with an acute disease course that may not be a good representative of ocular diseases with chronic and progressive neurodegeneration such as glaucoma. Larger randomized clinical trials are still needed in the future to verify the neuroprotective effects and evaluate possible adverse events of EPO in humans with glaucoma.

## Figures and Tables

**Figure 1 ijms-23-16038-f001:**
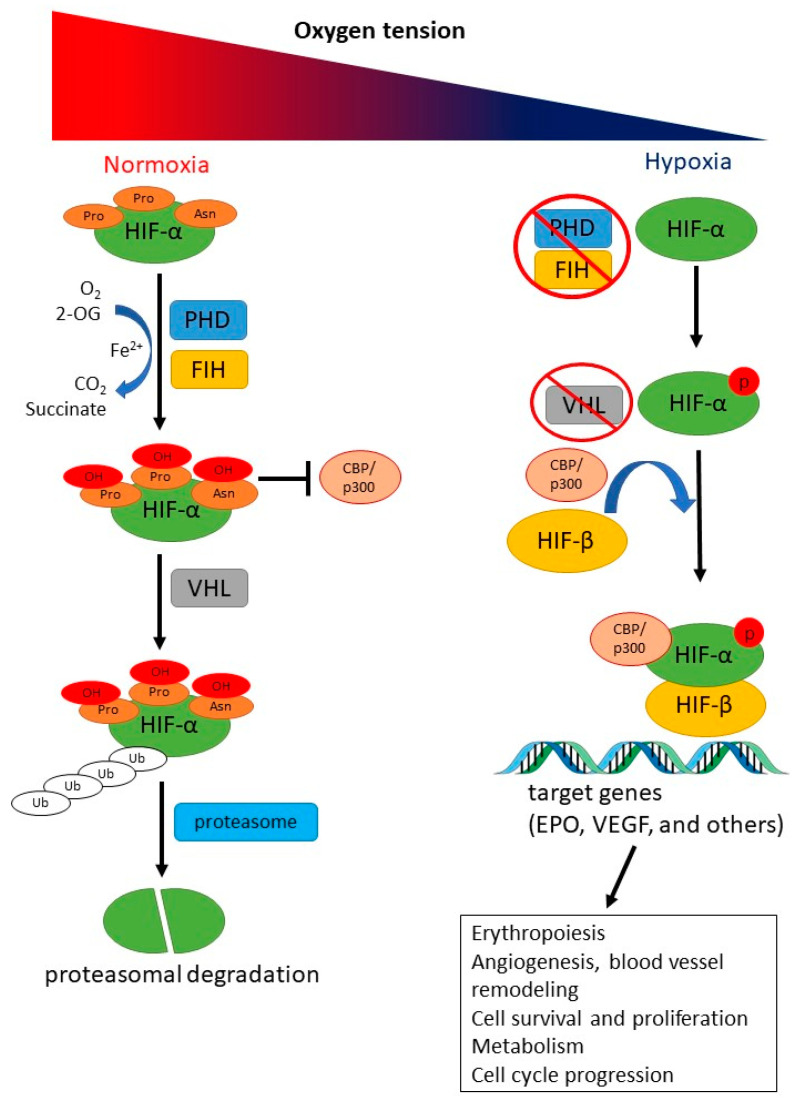
Oxygen-dependent regulation of hypoxia-inducible factor (HIF) function and erythropoietin (EPO) gene expression. The hypoxia-inducible factor (HIF) plays a key role in the hypoxic response through modulation of downstream genes which regulate vital biological processes such as angiogenesis, cell survival and proliferation, metabolism, erythropoiesis, and cell cycle progression. In normoxic status, prolyl hydroxylase domain (PHD) and protein factor inhibiting HIF-1 (FIH) proteins catalyze the hydroxylation of specific proline residues (position 402 and 564) and asparagine (position 803) in the HIF-α subunit, respectively, by an oxygen (O_2_), 2-oxoglutarate (2-OG), and in a ferrous ion (Fe^2+^)-dependent manner. Hydroxylated asparagine blocks the recruitment of transcriptional coactivator CREB-binding protein (CBP)/p300. The hydroxylated HIF-α subunit is subsequently ubiquitylated by von Hippel–Lindau tumor suppressor (VHL)-E3 ubiquitin ligase complex and rapidly degraded by the proteasome. In hypoxia, the function of PHD and FIH proteins is blocked. Stabilized HIF-α subunit is phosphorylated and translocates to the nucleus where it forms a functional heterodimer with the HIF-β subunit. The HIF-α/β heterodimers then bind to the hypoxia response elements (HRE) of the target genes (such as EPO, and VEGF) and increase gene transcription in the presence of transcriptional coactivators CBP/p300.

**Table 1 ijms-23-16038-t001:** Studies concerning the local or systemic administration of EPO toward glaucomatous optic neuropathy and RGC injury models.

Injury Model	Study Design	Methods	Sample Size	Authors	Year	Main Results and Remarks
Mechanical optic nerve injury						
Optic nerve transection	Intervention study	Groups: (1) Axotomy + vehicle (2) Axotomy + 0.5 U rhEPO (3) Axotomy + 1 U rhEPO (4) Axotomy + 2 U rhEPO (5) Axotomy + 4 U rhEPO (6) Axotomy + 8 U rhEPO (7) Axotomy +2 U rhEPO + 0.1 mM Wortmannin Intravitreal injections (IVI) on days 0, 4, 7, and 10 after optic nerve transection	25 adult Sprague–Dawley rats	Weishaupt et al. [51]	2004	EPO neuroprotection followed a bell-shaped dose–response curve without toxic effects at high concentrations. EPO-induced Akt phosphorylation and survival-promoting EPO effects were completely abolished by inhibition of PI-3-kinase.
Optic nerve transection	Comparative study	Comparison between tg21 mice versus wild-type mice 14 days after axotomy	5–6 animals per strain for tissue processing	Kilic et al. [52]	2005	RGCs viability: tg21 mice: 61.4 ± 20.6%; wild-type mice: 21.3 ± 7.6%. EPO protects RGCs against delayed degeneration via ERK-1/-2 signaling pathway.
Optic nerve transection	Intervention study	Optic nerve transection: (1) IVI of 5 units of EPO (2) IVI of 10 units of EPO (3) IVI of 25 units of EPO (4) IVI of 50 units of EPO (5) IVI of PBS Optic nerve transection + peripheral nerve grafting: (1) IVI of 25 units of EPO (2) IVI of PBS	58 PVG hooded rats	King et al. [53]	2007	A single IVI of EPO significantly increased the RGC somata survival (5, 10 units) and RGC axon survival (10, 25 units). A small proportion of axons penetrated the transection site and regenerated up to 1 mm into the distal nerve after intravitreal administration of EPO. IVI of EPO (25 units) doubled the number of RGC axons regenerating along a length of peripheral nerve grafted onto the retrobulbar optic nerve at 4 weeks.
Optic nerve crush	Intervention study	Groups: (1) no treatment (2) single intraperitoneal (i.p.) injection of 20 mg EPO-loaded PLGA/PLA microspheres (corresponding to 24,000 IU/kg EPO) (3) single i.p. injection of 20 mg blank PLGA/PLA microspheres (4) i.p. injection of EPO (3000 IU/kg BW) every week (5) i.p. injection of 0.01 M PBS every week	7 adult Sprague– Dawley rats/each group	Rong et al. [54]	2011	The release of EPO-loaded microspheres could last for at least 60 days in an in vitro study. A single dose of EPO-loaded microspheres showed a significant neuroprotective effect which was similar with multiple doses of EPO solution.
Optic nerve crush	Randomized, controlled, animal experiment	Groups: (1) sham-surgery group (2) IVI of 3.5 µL (6 U) EPO (3) IVI of 3.5 µL PBS IVI for 4 times: immediate (day 0), days 3, 6, and 9	72 adult Sprague–Dawley rats	Tan et al. [55]	2012	Compared with the PBS group, RGC densities in regions corresponding to the 1/6, 3/6 and 5/6 ratios of the retinal radius were significantly increased in the EPO group. EPO significantly increased growth associated protein(GAP)-43 protein levels in the retina after crush injury. EPO protects injured RGCs and promotes axonal regeneration by upregulating GAP-43 expression.
Episcleral vessel cautery-induced glaucoma model						
Episcleral venous cautery	Intervention study	Groups: (1) unoperated controls (2) episcleral vessel cautery only (EVC) (3) EVC + IVI of normal saline (EVC-NS) (4) EVC + IVI of recombinant rat EPO (EVC-EPO)	29 adult Sprague–Dawley rats (11 controls, 4 EVC, 5 EVC-NS, 9 EVC-EPO)	Tsai et al. [56]	2005	RGC counts: control: 12,619 ± 310; EVC: 9116 ± 273; EVC-NS: 9489 ± 293; EVC-EPO: 11,212 ± 414. A single IVI of 200 ng dose of EPO appears to have a protective effect on RGC viability in an in vivo rat model of glaucoma. The Con A Lectin staining did no show any gross vascular changes at 21 days in the EPO-treated eyes
1. Argon laser photocoagulation of episcleral and limbal veins 2. Optic nerve transection	Intervention study	Groups: (1) IVI of PBS (2) IVI of 2 μl sEPOR (20 ng) (3) IVI of 2U Epoetin alpha at 0, 4, 7, and 11 days after the injury (4) i.p. injection of 5000 units (42 μg, 1 mL) Epoetin alpha/kg (5) i.p. injection of normal saline 24 h before and 30 min before the injury * Chronic glaucoma model:(1)–(5); Optic nerve transection model:(4), (5)	adult Sprague–Dawley rats (*n* = 8/experiment group)	Fu et al. [57]	2007	Expression of EPO and EPOR proteins were increased significantly 2 weeks after ocular hypertension. sEPOR may neutralize the endogenous EPO to exacerbate the ocular hypertension injury, suggesting that the endogenous EPO/EPOR system may participate in the intrinsic recovery mechanisms after ocular hypertension. Both topical and systemic administration of rhEPO has significant neuroprotective activity on the survival of RGCs with no effect on IOP.
Coagulation of episcleral veins	Intervention study	Groups: (1) subconjunctival injection of 1000 IU of rhEPO (2) subconjunctival injection of equal volume of saline	26 Wistar Hannover albino rats	Resende et al. [58]	2018	In the ERG examinations, both flash and flicker b-wave amplitudes showed a significantly better recovery in the treated group. Compared with the control group, the treated group presented a statistically thicker retina at 21 days after glaucoma induction.
Model of anterior chamber cannulation and IOP elevation						
Cannulated anterior chamber (AC) and raised IOP to 120 mmHg for up to 60 min	Intervention study	Groups: (1) IVI of 2 ng sEPOR (2) IVI of 20 ng sEPOR (3) IVI of saline just before the induction of ischemia. (4) i.p. injection of 5000 units/kg rhEPO (5) i.p. injection of normal saline 24 h before and just before the onset of ischemia (6) i.p. injection of 5000 units/kg rhEPO immediately after the onset of ischemia	61 Sprague–Dawley rats	Junk et al. [9]	2002	48 h after ischemia, EPO protein levels dropped to 52% of the control eyes. EPOR protein levels peaked at 72 h after ischemia and were 4-fold higher than in control animals. At 7 days of reperfusion, the ERG b-wave was significantly reduced in sEPOR treated group with a dose-dependent effect. Systemic administration of rhEPO before or immediately after retinal ischemia not only reduced histopathological damage but also promoted functional recovery as assessed by ERG. Exogenous EPO significantly diminished TUNEL of neurons in the ischemic retina, implying an antiapoptotic mechanism of action.
Cannulated AC and raised IOP to 70 or 15 (sham-operated group) mmHg for up to 60 min	Intervention study	Groups: (1) untouched control (2) sham-operated (3) acute elevated IOP (4) acute elevated IOP + immediate retrobulbar injection of 100 μL rhEPO (1000 units) (5) acute elevated IOP + immediate retrobulbar injection of 100 μL vehicle solution	75 Sprague–Dawley rats	Zhong et al. [59]	2008	An acute elevated IOP could result in the loss of RGCs. Retrobulbar injection of rhEPO significantly preserved the number of surviving RGCs(per mm^2^) than acute elevated IOP group and acute elevated IOP + vehicle treated group. The densities of EPO and EPOR expression in the RGCs layer increased after acute IOP elevation, especially in the acute elevated IOP + rhEPO retrobulbar injection group.
1. The IOP was raised to 120 mmHg for 55 min 2. Optic nerve compression for 10 s	Intervention study	Groups: (1) IVI of PBS (2) IVI of 2U EPO (3) IVI of 20 U EPO For EPO safety evaluation: (1) IVI of 5 μL BSS (2) IVI of 5 U EPO (3) IVI of 50 U EPO (4) IVI of 200 U EPO	Brown Norway rats (*n* = 9–21 in elevate IOP, and *n* = 6–8 in optic nerve compression; *n* = 6–7 for safety evaluation)	Jehle et al. [60]	2010	IVI of 20 U EPO significantly increased the survival of RGCs by 127 ± 31% after ischemia and 58 ± 13% after optic nerve compression. The 20 U EPO significantly improved postischemic amplitudes of the A and B waves. Both 2 and 20 U EPO shortened the B-wave peak time after ischemic injury. Neither the ERG parameters, nor the VEP, nor the number of RGC differed significantly after intravitreal injection of EPO (5, 50, and 200 units, *n* = 6–7) in healthy eyes.
Oligemia by bilateral common carotid artery occlusion (BCCAO)						
BCCAO	Intervention study	Groups: (1) sham-operated group (2) BCCAO only (3) BCCAO + IVI of 1 μL PBS 2 days later (4) BCCAO + IVI of 1 μL EPO (400 μg) 2 days later	30 adult Wistar rats (*n* = 5 in sham and BCCAO only group)	Carvalho et al. [61]	2018	Treatment with EPO preserved the retinal thickness, the number of RGCs, and the organization of the layers compared to that in the other groups that underwent BCCAO surgery. The neuroprotective effect of EPO could be related to the down-modulation of glial reactivity.
BCCAO	Intervention study	Groups: (1) sham-operated+ intranasal saline (2) BCCAO + intranasal saline (3) BCCAO + intranasal rhEPO (50 U/100 g/week for 8 weeks)	90 Sprague–Dawley rats (*n* = 30/group))	Zhou et al. [62]	2020	rhEPO provided effective protection against deficits of spatial learning and memory in rats with chronic cerebral ischemia (CCI). rhEPO treatment markedly reduced the latency and elevated the amplitude of the P1 wave compared with the BCCAO + saline treatment group. rhEPO significantly preserved the thickness of cerebral cortex, the thickness of retina, the number of neurons in hippocampus CA1 area, and the number of RGCs in rats with CCI.
Mouse strain prone to develop glaucoma						
The DBA/2J mice spontaneously develop glaucomatous loss of RGCs with aging.	Intervention study	Groups: (1) i.p. injection of BSA (2) i.p. injection of memantine (70 mg/kg/wk) (3) i.p. injection of rhEPO-α (3000 U/kg/wk) (4) i.p. injection of rhEPO-α(6000 U/kg/wk) (5) i.p. injection of rhEPO-α(12,000 U/kg/wk)	294 DBA/2J mice and 91 C57BL/6J mice	Zhong et al. [63]	2007	Systemic administration of rhEPO (3000, 6000, and 12,000 U/kg/wk) promoted RGC survival in DBA/2J glaucomatous mice without affecting IOP. EPO effects were similar to those of memantine, a known neuroprotective agent.
Molecule-induced retinal toxicity						
1. Glutamate and nitric oxide toxicity 2. Cannulated AC and raised IOP to 120 mmHg for 60 min	Intervention study	*. Glutamate: (1) EPO (0.15, 0.5, and 1.5 U/mL) for 12 h + low concentration glutamate (50 and 100 μM) for 48 h (2) EPO (0.15, 0.5, and 1.5 U/mL) for 12 h + high concentration glutamate (1 and 10 mM) for 24 h *. Nitric oxide: EPO (1.5 U/mL) for 12 h + SNP(10, 100, and 500 μM) for 6 h *. Immunohistochemical analysis: right eye (operated eye), left eye (sham operated eye)	1. RGCs isolated from neonatal Wistar rats 2. Wistar rats for immunohistochemical analysis	Yamasaki et al. [64]	2005	EPOR was expressed in primary cultured RGCs, and RGCs in the normal and ischemic rat retina. Pretreatment with EPO for 12 h protects RGCs from glutamate and nitric oxide-induced cytotoxicity and reverses the Bcl-2 expression in a dose-dependent manner.
NMDA, TFW, and inflammatory toxicity	Intervention study	(1) NMDA (20–500 μM) + 1, 10, or 100 ng/mL of EPO (2) TFW+ 1, 10, or 100 ng/mL of EPO (3) Inflammatory (TNF-α at 12.5–50 ng/mL) + 1, 10, or 100 ng/mL of EPO Time-course effects of EPO exposure: (1) EPO at 8 h before (2) EPO at 4 h before (3) EPO at 0 h (3) EPO at 4 h after (3) EPO at 8 h after	RGCs isolated from adult Wistar rats	Chang et al. [65]	2013	EPO provides different degree of neuroprotective effect depending on the type of neurotoxicity, RGC cell size, and the timing of treatment. Inhibitors of signal transduction and activators of transcription such as STAT-5, MAPK/ERK, and PI3K/Akt impaired the protective effect of EPO on RGCs exposed to different insults
NMDA toxicity	Randomized intervention study	Groups: (1) negative control (2) IVI of 80 nmoles NMDA80 (3) IVI of 80 nmoles NMDA80 + 10 ng EPO (4) IVI of 80 nmoles NMDA80 + 50 ng EPO (5) IVI of 80 nmoles NMDA80 + 250 ng EPO	125 Wistar rats	Cheng et al. [66]	2020	An intravitreal injection of 50 ng EPO showed a significantly higher survival rate of the total, medium, and small RGCs after NMDA-mediated neurotoxicity. EPO co-administered with NMDA downregulated the generation of pro-apoptotic proteins, such as μ-calpain, caspase 9, and Bax in ganglion cell layer. EPO reversed the damage to bipolar cell axon terminals in the inner plexiform layer induced by NMDA.
Microbeads-induced glaucoma model						
Elevated IOP induced by intracameral injection of microbeads	Intervention study	Groups: (1) IVI of EPO-R76E loaded PLGA particles (2) IVI of EPO-R76E loaded PPS microspheres one day after elevation of IOP	Mice	Rex et al. [67]	2022	Both PLGA-EPO-R76E and PPS-EPO-R76E particles showed neuroprotective effects in microbeads-induced glaucoma mice model through activation of the NRF2/ARE pathway.

* BCCAO, bilateral common carotid artery occlusion; BSS, balanced salt solution; CCI, chronic cerebral ischemia; EPO, erythropoietin; EVC, episcleral vessel cautery; GAP, growth associated protein; IOP, intraocular pressure; i.p., intraperitoneal; IVI, intravitreal injection; NMDA, N-Methyl-D-aspartic acid; PLGA/PLA, poly lactic-co-glycolic acid/poly lactic acid; PBS, phosphate buffer solution; PPS, poly(propylene sulfide); NRF2/ARE, nuclear factor erythroid 2-related factor 2/antioxidant response element; RGC, retinal ganglion cell; SNP, sodium nitroprusside; TFW, trophic factor withdrawal; TNF-α, Tumor Necrosis Factor-α; VEP, visual evoked potentials.

## Data Availability

Not applicable.

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
