# Peer review of "The Latest Evidence of Erythropoietin in the Treatment of Glaucoma"

_ijms, 2022, doi:10.3390/ijms232416038_

Round 1
Reviewer 1 Report
I would like to apologize with editor and authors for the delay in my response. I have had problem with the mdpi account that was linked to my previous institutional email address (that is now not functional). I hope that the problem is now solved.
The authors collected and discussed an extensive amount of information about the use of erythropoietin in the treatment of glaucoma. the review is clear and well organized in paragraphs. The table is also very clear and exhaustive.
Here some minor suggestions:
- the paragraph in lines 66-80 is not clearly linked to the EPO expression mechanism, please clarify explicitly. A figure with a schematic representation of the EPO expression and/or EPOR activity could help
- the paragraph in lines 257-263 should be moved in the discussion (now is the paragraph related to Microbeads induced glaucoma model
- some moderate english corrections are needed (e.g. line 39 " to BE expressed", line 73 "The hydroxylated HIF-α IS subsequently ubiquitylated by von Hippel-Lindau tumor suppressor"
- clarify some acronyms (e.g. line 45 Hb; line 153 PLGA and PPS)
- clairfy the role of p300 (line 80)
-clarify the informations derived from TUNEL (line 193)
- references in lines 280 and 326 are of different size
Reviewer 2 Report
The authors provide an interesting and clear summary of the role of EPO as it relates to glaucoma. In particular, the summary of table 1 is very attractive, detailed and easy to understand for the non-specialist.
Reviewer 3 Report
Lu et al present a comprehensive review of the literature regarding EPO and glaucoma, mainly focused on animal models. I usually have comments and suggestions to offer, but it appears that the work is fine in its present form.
